

# The size of signal detection and emission organs in a synchronous firefly: sexual dimorphism, allometry and assortative mating

Tania López-Palafox[1], Rogelio Macías-Ordóñez[2] and
Carlos R. Cordero[3]

[1] Posgrado en Ciencias Biomédicas, Instituto de Ecología, Universidad Nacional Autónoma de México, Mexico City, Mexico
[2] Instituto de Ecología A.C., Xalapa, Veracruz, México
[3] Departamento de Ecología Evolutiva, Instituto de Ecología, Universidad Nacional Autónoma de México, Mexico City, México

Corresponding author
Carlos R. Cordero,
cordero@ecologia.unam.mx

## ABSTRACT

The size of the organs responsible for emitting and detecting sexual communication signals is a likely target for selection. Communication via bioluminescent signals in synchronous fireflies is a promising model to test hypotheses regarding differences between males and females in the effect of the size of signal emission and detection organs on fitness components. Synchronous firefly species congregate in large numbers during the mating season, displaying bioluminescent signals aimed at potential mates during relatively short nightly periods. Operational sex ratios are male-biased and, thus, the so-called typical sex roles (indiscriminate males and choosy females) are expected to evolve. We studied the synchronous firefly *Photinus palaciosi*, a species that during the mating season congregates in forests of central Mexico offering a magnificent natural show that attracts numerous tourists. *P. palaciosi* females have reduced wings (brachyptery) and cannot fly. Our field study tested the hypothesis that the male-biased operational sex ratio and the short daily mating period result in strong male-male competition that selects for males with larger lanterns and larger eyes, and against male mate choice, whereas female-female mate competition is absent and, thus, no selection on lantern or eye size is expected. Even though lantern, eye or body size do not predict the probability of being found in copula for either sex, sexual dimorphism in these features, along with allometric slopes of lantern size and assortative mating in terms of relative lantern size, support not only the hypothesis of intense sexual selection among males, but the possibility of subtle mechanisms of sexual selection among females. Trade-offs between investment in signaling (lanterns) versus detection (eyes) structures, or with pressures different from sexual selection such as those imposed by predators, are also likely to be important in shaping the evolution of sexual signaling in these fireflies.

## INTRODUCTION

Communication between males and females is a fundamental element of the mating biology of most animals (*Darwin, 1871*; *Maynard Smith & Harper, 2003*; *Rosenthal, 2017*). There is a great variety of organs, newly evolved or specialized via sexual selection, for the emission and reception of sexual signals (*Darwin, 1871*; *Rosenthal, 2017*; *Elgar, Johnson & Symonds, 2019*). A fascinating example of sexual communication involving vision is that of nocturnal fireflies (*Lloyd, 1979*; *Lewis, 2016*). In these insects, adults of many species possess an organ specialized for the emission of light known as lantern. Typically, males fly searching for females, emitting species- and sex-specific flashing patterns that are involved in mate choice, while females emit glows or flashes in response (*Lloyd, 1979*; *Lewis & Cratsley, 2008*; *Lewis, 2016*; *Stanger-Hall et al., 2018*). If a successful dialog is established, the male alights, contacts the female and a close-range courtship ensues (*Lewis & Cratsley, 2008*; *Stanger-Hall et al., 2018*).

Although female choice has been related to the duration and pulse pattern of male flashes (*Lewis, Cratsley & Demaris, 2004*; *Lewis & Cratsley, 2008*), it is reasonable to propose that signal intensity, and thus the sizes of the lantern and of the eyes are also important traits influencing the efficiency of sexual communication in fireflies (*Vencl & Carlson, 1998*; *Cratsley & Lewis, 2003*, *2005*; *Demary, Michaelidis & Lewis, 2006*; *Lau & Meyer-Rochow, 2006*). Larger lanterns may increase signal transmission distance (*Demary, Michaelidis & Lewis, 2006*), whereas larger eyes are correlated with smaller interommatidial angles that may help improve visual resolution and distance perception (*Lewis, Cratsley & Demaris, 2004*), as well as capture more photons thus helping vision in low-light environments (*Warrant & Dacke, 2011*; *Stanger-Hall et al., 2018*). Somewhat surprisingly, studies on the relationship between signal-emission organ size and mating success in fireflies are scant and their results are inconsistent. While some studies in non-synchronic fireflies detected an effect of male lantern size on mating success and female responses in a high-density population of *Photinus pyralis* (*Vencl & Carlson, 1998*) and in *P. ignitus* (*Cratsley & Lewis, 2003*, *2005*), another study found that female mating decisions in the non-synchronic *P. greeni* are not influenced by lantern size (*Demary, Michaelidis & Lewis, 2006*). The effects of signal-detection organ (eyes) size on fitness components of both sexes, and of signal emission organ size on female fitness components have not been studied in fireflies (but see *Cratsley & Lewis, 2005*).

In this context, the static allometry of sexual characters (i.e., variation of their relative size among the size range of adults in the population) involved in courtship may be of particular interest. Selection is expected to optimize relative size of any organs depending on their reproductive payoff given each individual's body size. Unlike sexually dimorphic characters involved in intra-sexual agonistic signaling (usually among males) in which larger individuals are expected to show disproportionally larger traits (i.e., positive allometry), the size of sexually dimorphic signaling characters involved in courtship are expected to be more frequently either proportional to body size (i.e., isometry) or even relatively smaller in larger individuals (i.e., negative allometry) (*Eberhard et al., 2018*). The reason to expect these patterns are diverse, but in general the

payoff of relatively larger traits for larger individuals is high when they are involved in agonistic interactions in which body size is a good predictor of fight outcome, thus selecting for conflict resolution prior to engaging in costly or dangerous fights.

In male-female reproductive interactions, however, the conditions are much more variable. In many cases selection for "honest" signals accurately reflecting male body size would result in isometry, while in others relatively smaller organs in large males (i.e. negative allometry) result in high payoffs if male quality is not directly related to body size (*Eberhard et al., 2018*). Furthermore, the allometry of sexual organs involved in receiving rather than emitting signals, as well as sexual differences in allometry, have seldom been explored.

Synchronous flashing fireflies are good subjects to study hypotheses on the effects of the size and allometry of the organs involved in sexual communication on fitness components because in these species the density of signaling males is very large, nightly mating periods are short and the operational sex ratio is male biased, resulting in intense competition between males, likely absence of female-female mate competition, and plenty of opportunities for female choice (*Lloyd, 1979*; *Lewis, 2016*). We can hypothesize that intense male-male competition selects for males with larger lanterns that increase signal transmission distance. Selection would also favor males with larger eyes that increase the amount of photons captured in the night and improve the detection distance of the usually faint glows produced by the relatively scarce females. Larger males, but not females, may have a mating advantage over small males due to direct selection on body size (e.g., if larger male size is advantageous when several males alight simultaneously and court a female; *Thornhill & Alcock, 1983*) or correlative selection (e.g., if selection favors larger lantern size and this measure is positively correlated with body size).

In contrast to males, females would be selected to emit flashes of just enough intensity to be perceived by the males, thus no selection for increased lantern size is predicted. In fact, it is even possible that females are selected to produce less intense flashes or glows not only reducing costs, but as a female choice mechanism that allows being detected only by particularly sensitive males, an ability they could inherit to their male offspring (*Eberhard, 1996*). On the other hand, selection for an increase in eye size should be relaxed in females due to the fact that they are the limiting sex and the large number of potential mates encountered every night, although larger eyes could be advantageous if they increase the ability to detect and respond to high quality males. Since larger signaling and receiver structures should benefit larger males despite their costs, but would represent also higher costs but no additional benefits for larger females, we could expect steeper allometric slopes for lantern and eye size in males than in females. In this paper, we present the results of a field study aimed at testing some of the predictions derived from these hypotheses.

We studied the synchronous Mexican firefly *Photinus palaciosi* (*Zaragoza-Caballero, 2012*; *Zaragoza-Caballero et al., 2020*). Using field-collected data, we estimated the relationship between the probability of being found in copula and signal emission (lantern) and signal detection (eyes) organ size, and used these data to test the following predictions: (1) Males with larger lanterns and eyes have higher probability of being found in copula, whereas lantern and eye size in females are not related to their mating

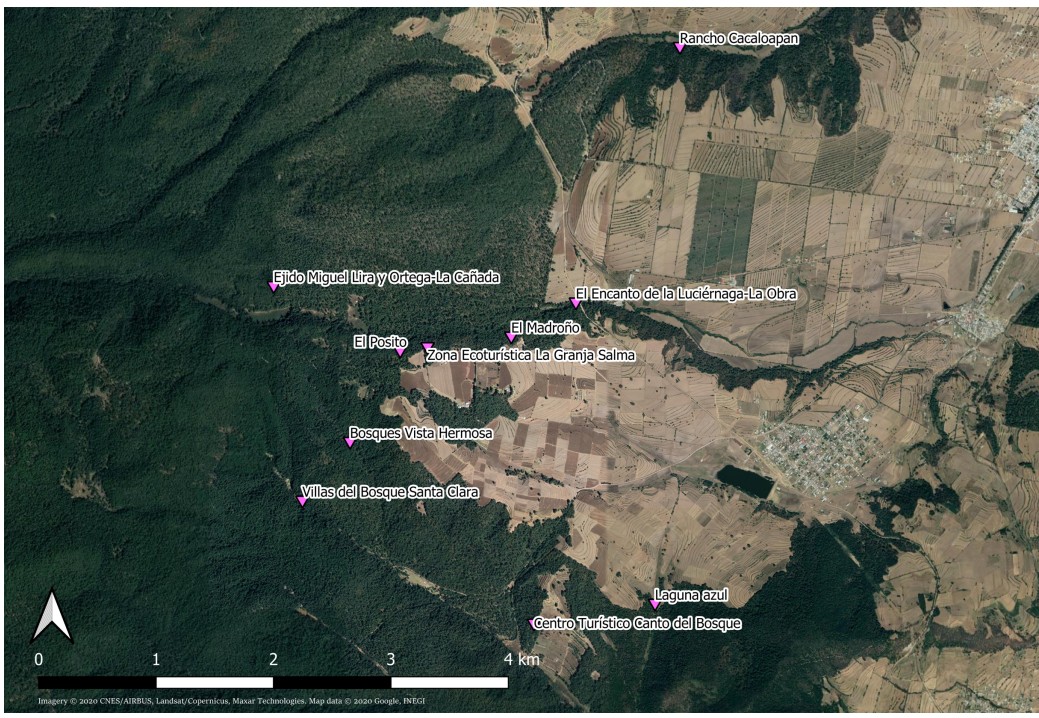

**Figure 1 Collection sites of *Photinus palaciosi* in the forest of the municipality of Nanacamilpa de Mariano Arista (Tlaxcala state, México).** All sites were administrated by private eco-touristic businesses. Imagery ©2020 CNES/AIRBUS, Landsat/Copernicus, Maxar Technologies. Map data ©2020 Google, INEGI. Downloaded August 27, 2020.           

probability. (2) Larger males have a higher probability of being found in copula. (3) Males have larger lanterns and larger eyes than females. (4) Lantern and eyes allometric slopes are higher in males than in females, but not higher than 1 since they are not involved in male-male agonistic interactions. Finally, (5) there is no assortative mating for lantern, eye or body size due to the lack of intra-female sexual selection or male mate choice under highly male competitive conditions.

## MATERIALS AND METHODS

### Species studied

*Photinus palaciosi* lives in pine-oak-fir forests of central Mexico, in the states of Estado de México, Puebla and Tlaxcala, and its reproductive season goes from June to the beginning of August. Mate searching, courtship and mating occur during approximately ninety minutes every night starting around 20:30 h, although heavy rainfall prevents flying activity. In the study site (Fig. 1), the municipality of Nanacamilpa de Mariano Arista (Tlaxcala state, México), thousands of males congregate under the canopy of the forest during this period, flying in search of females, frequently synchronizing their flashing and providing a magnificent show that attracts numerous tourists (*Acle Mena et al., 2018*). The females cannot fly because they are brachypterous (i.e., their wings are extremely reduced; Fig. 2) (*Zaragoza-Caballero, 2012*), and they remain stationary in herbs at heights <60 cm and glow infrequently during the mating period. The number of sexually

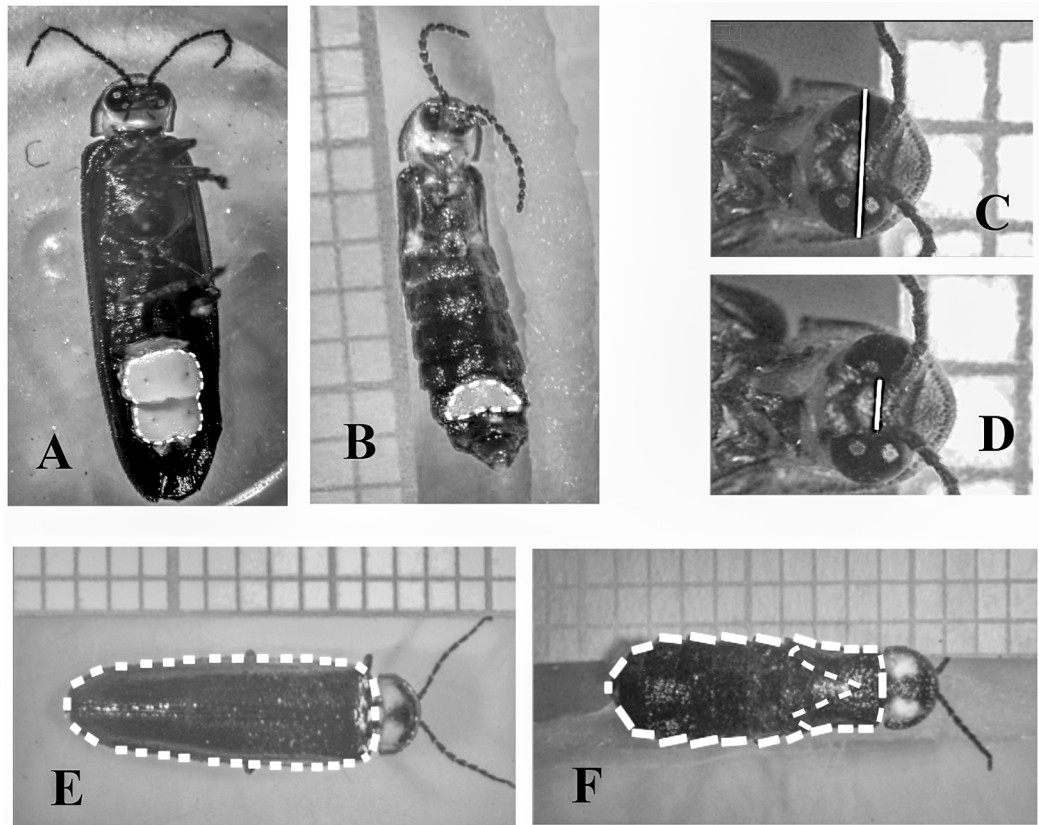

**Figure 2 Measurements performed in males and females of *Photinus palaciosi*.** (A) Male lantern size area, (B) female lantern size area, (C) maximum eye span, (D) interocular space, (E) male elytra area, (F) female body area and elytra area.

receptive females every night is much smaller than that of males and thus the operational sex ratio (OSR) is male biased (T. López-Palafox & C. Cordero, 2016–2019, personal observation).

## Sample collection

Samples of males and females found in copula or solitary were collected simultaneously by a team of three researchers during the daily mating period (20:30–22:00 h GMT-5) in the middle of the 2016 reproduction season (between June 29 and July 14). Signalling males were collected with an entomological net and solitary females and mating couples by hand. Individual mating couples and solitary individuals were kept in Eppendorf vials with absolute alcohol. Captures were made in 10 places (one per night) within a continuous forests in the municipality of Nanacamilpa de Mariano Arista (Table 1; Fig. 1). Our collection was made under the SEMARNAT (Mexican Government) permit SGPA/DGVS/06292/16.

## Measurement of phenotypic traits

We obtained three photographs of each firefly (dorsal view, ventral view and a close up of the eyes) with a digital camera (Model T3i; Canon™, Tokyo, Japan) mounted on a

**Table 1 Collection sites of *Photinus palaciosi* in the forest of the municipality of Nanacamilpa de Mariano Arista (Tlaxcala state, México).**

| Site | Date | Solitary females | Females in copula | Solitary males | Males in copula |
|------|------|------------------|-------------------|----------------|-----------------|
| 1. Centro Turístico Canto del Bosque | 29/06 | 10 | 1 | 15 | 1 |
| 2. Laguna azul | 30/06 | 6 | 7 | 9 | 7 |
| 3. El Madroño | 1/07 | 9 | 2 | 12 | 2 |
| 4. Zona Ecoturística La Granja Salma | 2/07 | 8 | 5 | 11 | 5 |
| 5. El Posito | 3/07 | 6 | 4* | 11 | 4 |
| 6. Rancho Cacaloapan | 4/07 | 2 | 5* | 8 | 5 |
| 7. Ejido Miguel Lira y Ortega-La Cañada | 5/07 | 7 | 5 | 2 | 5 |
| 8. Villas del Bosque Santa Clara | 6/07 | 3 | 3 | 0 | 3 |
| 9. Bosques Vista Hermosa | 8/07 | 0 | 8 | 1 | 8 |
| 10. El Encanto de la Luciérnaga-La Obra | 14/07 | 0 | 4 | 0 | 4 |

Note:
All sites were administrated by private eco-touristic businesses. Collection dates and numbers of females and males captured when solitary or in copula in each site are provided. Two copulating females (*) were not included in the analyses because the posture they had after fixation in alcohol prevented obtaining correct measurements.

disection microscope (Model SZH10; Olympus™, Tokyo, Japan). The phenotypic measurements were obtained with the NIH ImageJ open access software (National Institutes of Health USA, http://rsb.info.nih.gov.ij/). We estimated lantern size by measuring the area covered by the lantern in the ventral-view photographs (Fig. 2). Eye size was estimated as the squared difference between maximum eyespan and interocular space (i.e., approximately the sum of the maximum diameter of both eyes) in the eyes close-up photographs (Fig. 2). The reason to square this length was to obtain a variable that would covariate linearly with the rest of the body traits since they were all area measurements. Body size was estimated as the area covered by the elytra of the males in the dorsal view photographs, while in females it was estimated as the area covered by the thorax and the abdomen (Fig. 2). The area covered by the reduced female elytra was also measured to document brachyptery quantitatively, and as an additional proxy of body size (see below).

## Statistical analyses

For predictions 1 and 2, we constructed separate binomial generalized linear models with logit link function for each sex, using mating status (0 = captured alone, 1 = captured mating) as the binary response variable, and lantern size, eye size (prediction 1), and body size (prediction 2) as explanatory variables, as well as their second order interactions. We simplified the models using backwards stepwise simplification, removing each explanatory variable in order of increasing significance and testing the effect of removing that variable with a chi-squared likelihood ratio test until only terms whose removal leads to worsening of the model remained (*Crawley, 2013*). A second set of models was constructed and simplified using relative lantern size and relative eye size in order to rule

out any effect of colineality among explanatory variables. Relative sizes were estimated dividing trait (lantern or eye) size by body size. A third set of models was constructed and simplified for females in order to rule out the effects of body condition using elytra area as a proxy of body size to construct lantern and eye relative size as explanatory variables. Unlike wing area, abdomen volume (and thus measured abdomen area) may vary through adulthood depending on body condition, which in turns may depend on water, muscle and fat reserves, and egg load in the case of females (*Moya-Laraño et al., 2008*). Egg load has been shown to be associated to female weight in a congeneric species of brachypterous females (*Wing, 1989*).

For prediction 3, we performed a simple linear model for each morphological trait (body size, elytra size, lantern size, lantern relative size, eye size and eye relative size) as dependent variable, and sex as the only explanatory variable. Although we had no prediction for body size sexual dimorphism, and elytra size dimorphism is so large that does not require a statistical test, we included these variables in the analyses for descriptive purposes. For prediction 4 we estimated the slope of log–log relationships between lantern size or eye size and body size using ordinary least squares (*Kilmer & Rodríguez, 2016*). As described above for prediction 1, a second set of analyses was performed for females using elytra area instead of body size in order to rule out effects of body condition. In order to compare slopes statistically, 95% confidence intervals were generated for each allometric slope using bootstrap resampling with 10,000 randomizations (*Crawley, 2013*)

For prediction 5, we only used individuals captured in copula and tested assortative mating using linear models relating morphological traits (lantern size, lantern relative size, eye size, eye relative size or body size) of males and their respective female mates. As described above for predictions 1 and 3, female elytra area was included as proxy of female body size in order to rule out effects of body condition.

We carried out these analyses in R software, version 3.6.3 (*R Core Team, 2020*) using the R Studio interface (*R Studio Team, 2016*). The script used for analyses and the databases with all data can be found as Supplemental Material (Files S1–S3).

## RESULTS

### General observations
We sampled 93 females (51 solitary and 42 in copula) and 113 males (71 solitary and 42 in copula). Two copulating females were not included in the analyses because the posture they had after fixation in alcohol prevented obtaining correct measurements.

### Phenotypic traits and the probability of capture during mating
We did not find support for predictions 1 or 2. Neither body size nor any of the morphological traits associated to signal emission (lantern size, lantern relative size) or reception (eye size or eye relative size), nor their statistical interactions, showed a significant association with the probability of being captured single or mating for either sex (Table 2).

**Table 2 Results of five models evaluating association between probability of being collected copulating rather than alone and trait size.**

| Fixed effects | β ± SE | Z | $P_Z$ |
|---|---|---|---|
| (A) Absolute size: males | | | |
| Body (elytra) | −0.087 ± 0.166 | −0.52 | 0.602 |
| Lantern | −0.517 ± 1.371 | −0.38 | 0.706 |
| Eyes$^2$ | 0.758 ± 0.827 | 0.92 | 0.360 |
| Body × Lantern | 0.029 ± 0.032 | 0.91 | 0.364 |
| Body × Eyes$^2$ | −0.005 ± 0.053 | −0.09 | 0.931 |
| Lantern × Eyes$^2$ | −0.149 ± 0.392 | −0.38 | 0.704 |
| (B) Absolute size: females | | | |
| Body (elytra) | −0.163 ± 0.278 | −0.59 | 0.557 |
| Lantern | −9.260 ± 7.539 | −1.23 | 0.219 |
| Eyes$^2$ | 0.119 ± 0.929 | 0.13 | 0.898 |
| Body × Lantern | 0.355 ± 0.299 | 1.19 | 0.235 |
| Body × Eyes$^2$ | −0.008 ± 0.046 | −0.17 | 0.864 |
| Lantern × Eyes$^2$ | 0.036 ± 1.149 | 0.03 | 0.975 |
| (C) Relative size: males | | | |
| Body (elytra) | 0.133 ± 0.316 | 0.42 | 0.673 |
| Lantern | 44.192 ± 98.546 | 0.45 | 0.654 |
| Eyes$^2$ | 62.479 ± 50.495 | 1.24 | 0.216 |
| Body × Lantern | −0.540 ± 2.664 | −0.20 | 0.839 |
| Body × Eyes$^2$ | −0.663 ± 0.800 | −0.83 | 0.408 |
| Lantern × Eyes$^2$ | −349.314 ± 364.564 | −0.96 | 0.338 |
| (D) Relative size: body females | | | |
| Body | 0.189 ± 0.270 | 0.70 | 0.484 |
| Lantern | 12.014 ± 147.610 | 0.08 | 0.935 |
| Eyes$^2$ | 19.705 ± 23.523 | 0.84 | 0.402 |
| Body × Lantern | −0.836 ± 5.693 | −0.15 | 0.883 |
| Body × Eyes$^2$ | −0.752 ± 0.804 | −0.93 | 0.350 |
| Lantern × Eyes$^2$ | −136.556 ± 312.171 | −0.44 | 0.662 |
| (E) Relative size: elytra females | | | |
| Elytra | −0.102 ± 0.254 | −0.402 | 0.688 |
| Lantern | −51.635 ± 41.787 | −1.24 | 0.217 |
| Eyes$^2$ | −1.407 ± 5.306 | −0.26 | 0.791 |
| Body × Lantern | 1.444 ± 1.486 | 0.97 | 0.331 |
| Body × Eyes$^2$ | −0.092 ± 0.177 | −0.2 | 0.605 |
| Lantern × Eyes$^2$ | 20.733 ± 20.471 | 1.013 | 0.311 |

Note:
Absolute trait size in males (A), females (B), trait size relative to body (elytra) size in males (C), trait size relative to body size in females (D), and trait size relative to elytra size in females (E) in the synchronous firefly *Photinus palaciosi*. Parameters from initial models are presented since backwards stepwise simplification resulted in removal of all explanatory variables in all models.

## Sexual dimorphism

Males are significantly larger than females, and as could be expected given the brachypterous morphology of females, male wing area is also significantly larger (Table 3).

**Table 3 Comparison of the morphological measurements of male and female *Photinus palaciosi* fireflies.**

| | Females | Males | Linear model $p$ ($F_{1,204}$) |
|---|---|---|---|
| Body area (mm$^2$) | 20.43, 20.78 (11.66–40.69) | 30.92, 30.73 (19.68–53.64)[a] | <0.001 (152.4) |
| Elytra area (mm$^2$) | 4.65, 4.90 (3.13–9.43) | 30.92, 30.73 (19.68–53.64)[a] | <0.001 (1493) |
| Lantern area (mm$^2$) | 0.74, 0.79 (0.41–2.04) | 3.52, 3.59 (2.14–6.08) | <0.001 (925.8) |
| Lantern relative size × 100[b] | 3.65, 3.91 (2.84–8.38) | 11.69, 11.70 (7.69– 15.09) | <0.001 (2095) |
| Eyes$^2$ (mm$^2$) | 3.29, 3.37 (0.54–8.77) | 1.56, 1.93 (0.50-6-14) | <0.001 (59.27) |
| Eyes relative size × 100[b] | 15.58, 17.10 (2.71-41.40) | 5.17, 6.51 (1.49– 25.58) | <0.001 (141) |

Notes:
[a] Elytra area was used as the main estimate of male body size.
[b] Relative size is multiplied by 100 in order to illustrate more decimal points.
Median, mean (minimum value – maximum value) are given for each trait.

**Table 4 Allometric slopes of lantern and eye size for females and males of the firefly *Photinus palaciosi*.**

| | Lantern (mm$^2$) | Eyes$^2$ (mm$^2$) |
|---|---|---|
| **Females** | | |
| Body area (mm$^2$) | *m = 0.56 [0.32, 0.78]* <br> *R$^2$ = 0.28, p < 0.001* | m = −0.04 [−0.51, 0.37] <br> R$^2$ = 0.01, p = 0.829 |
| Elytra area (mm$^2$) | *m = 0.67 [0.39, 0.94]* <br> *R$^2$ = 0.30, p < 0.001* | m = −0.15 [−0.68, 0.30] <br> R$^2$ = 0.01, p = 0.524 |
| **Males** | | |
| Body (=Elytra) (mm$^2$) | *m = 1.01 [0.91, 1.11]* <br> *R$^2$ = 0.74, p < 0.001* | m = 0.28 [−0.21, 0.77] <br> R$^2$ < 0.01, p = 0.297 |

Notes:
Two proxies of body size were used for females, body and elytra area, and one, body (=elytra) area, for males.
$m$ = allometric slope value [95% Bootstrap confidence interval]. Parameters of significant correlations in italics.

We found partial support for prediction 3. As predicted, males have significantly larger lanterns than females in terms of absolute and relative size. Interestingly however and contrary to our prediction, they have significantly smaller eyes than females in terms of absolute and relative size.

### Static allometry

As predicted, the slope of male lantern size allometry was significantly higher than the slope of female lantern size allometry, but not significantly higher than 1 (Table 4; Figs. 3A, 3C, and 3E). In contrast, eye size allometry was not significantly different from 0 in males or females, and its relative size varied widely along body size ranges in both sexes (Figs. 3B, 3D, and 3F). In other words, lantern relative size is constant along the male size range, but in females lantern relative size decreases somewhat towards larger individuals. On the other hand, eye size does not covary with body size in either sex.

### Assortative mating

Support for prediction 5 was also partial. We found no correlation between male and female traits in most of the variables (Table 5), but we did find a significant association

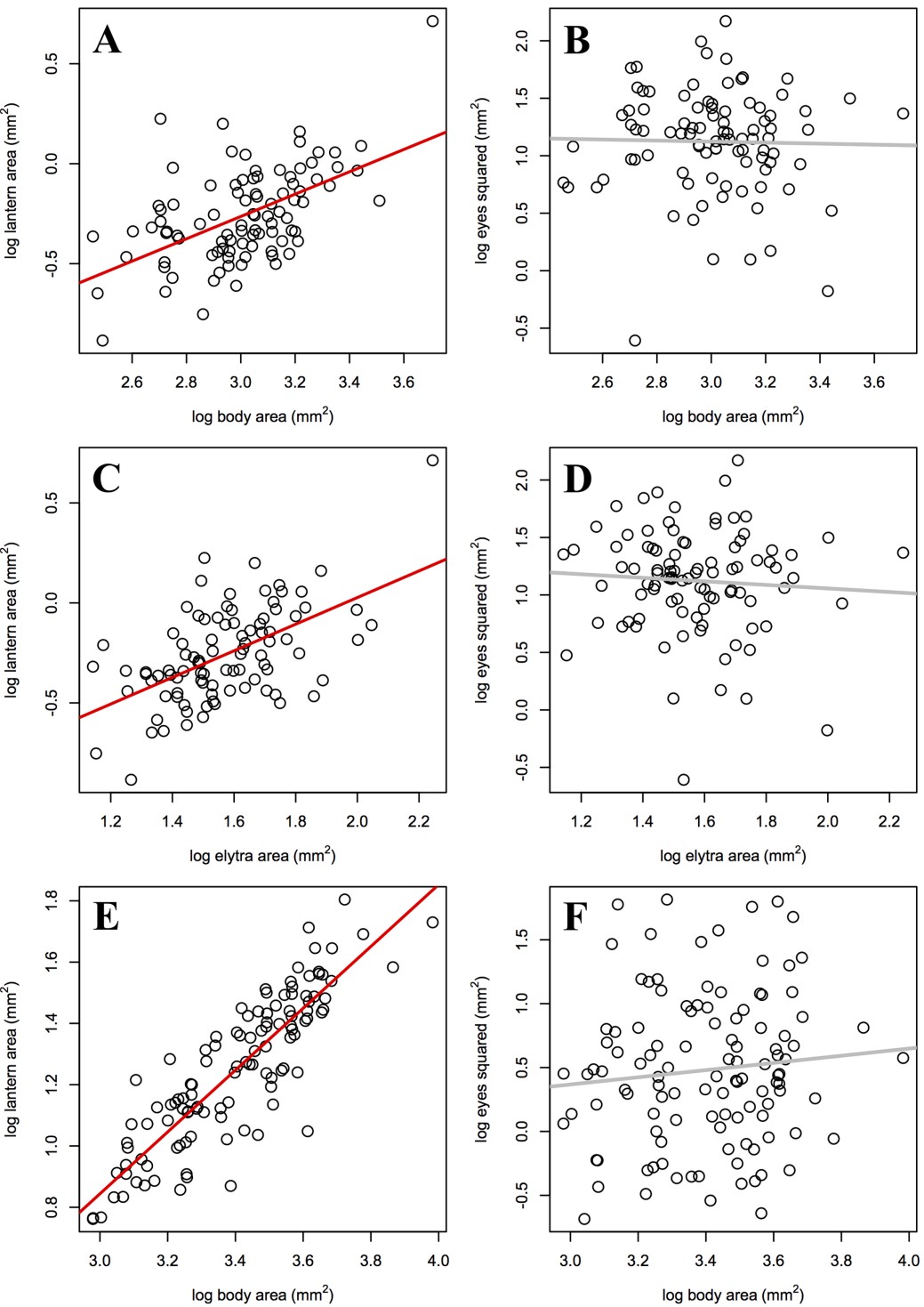

**Figure 3 Static allometry of the synchronous firefly *Photinus palaciosi*.** (A) Female lantern size relative to body size, (B) female eye size relative to body size, (C) female lantern size relative to elytra size, (D) female eye size relative to elytra size, (E) male lantern size relative to body (elytra) size, (F) male eye size relative to body (elytra) size. Lines in red represent slopes significantly higher than 0 and lines in gray represent slopes non significantly different from 0 (see Table 4 for statistical parameters).

**Table 5 Correlation of the morphological traits of males and females found in copula in *Photinus palaciosi* fireflies.**

| Male vs female trait | Absolute size | Relative size (female body) | Relative size (female elytra) |
|---|---|---|---|
| Body | $R^2 = 0.024$ | – | – |
| | $p = 0.851$ | – | – |
| Elytra | $R^2 = 0.008$ | – | – |
| | $p = 0.258$ | – | – |
| Lantern | $R^2 = 0.011$ | $R^2 = 0.021$ | $R^2 = 0.153$ |
| | $p = 0.236$ | $p = 0.176$ | $p = 0.006$ |
| Eyes$^2$ | $R^2 = 0.010$ | $R^2 = 0.020$ | $R^2 = 0.019$ |
| | $p = 0.443$ | $p = 0.676$ | $p = 0.642$ |

**Notes:**
Two versions of lantern and eyes relative size were calculated for females, one using body area and other using elytra area.
Significant correlations in italics.

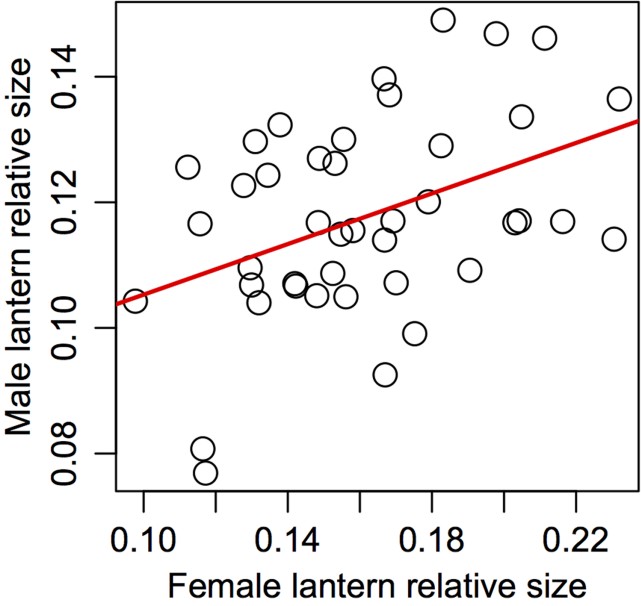

**Figure 4 Assortative mating in relative lantern size in the synchronous firefly *Photinus palaciosi*.** See Table 5 for statistical parameters.

between male and female relative lantern size when the effect of body condition was removed using the area of female elytra to estimate relative lantern size (Fig. 4).

## DISCUSSION

In this article we tested the hypothesis that in synchronous fireflies the male biased operational sex ratios and the short nightly mating period result in strong male-male competition that selects for males with larger signal emission (lantern) and signal detection (eyes) organs, as well as larger body sizes. On the other hand, since in these fireflies male mate choice and female-female competition for mates are expected to be absent, no selection on body and lantern size is expected in females, although intersexual selection

(female choice) could favor females with larger signal detection organs (eyes). We did not find support for the predictions that body size, lantern size or eye size would be associated with the probability of being found in copula. However we did find that males are not only larger than females, they have relatively larger lanterns but smaller eyes than females, even adjusting for sexual differences in body size. Furthermore, the allometric slope of lantern size is steeper in males than in females, but the allometric slope for eye size does not differ from 0 in either sex. Finally, and contrary to our predictions, there is some evidence of assortative mating in terms of lantern relative size.

Perhaps the best species to compare our results is *P. pyralis* (*Vencl & Carlson, 1998*), a species resembling *P. palaciosi* in that there is "intense competitiveness: aggregations of males regularly attain very high densities", sometimes resulting in several males attempting to mate with the same female (*Vencl & Carlson, 1998*), as we have observed in *P. palaciosi* (T. López-Palafox, Jaime Camacho & C. Cordero, 2017–2019, personal observations). In contrast to our findings, in *P. pyralis* the body size (elytral length) and lantern area of males were related to the probability of being found in copula. Interestingly, in this species larger males and males with larger lanterns were more successful when single males courted females (the most common case: 70% of all matings), but smaller males had an advantage when four or more males simultaneously courted a female "on foot" on her perch (12% of all matings). According to the authors, these contrasting effects "obscured" the global effect of elytral and lantern length on male mating success (*Vencl & Carlson, 1998*). When we collected many of the copulating pairs there was at least one additional male close to the copulating pair, unfortunately we did not make a record of this fact. However, a trade-off similar to that proposed by *Vencl & Carlson (1998)* may explain the lack of effects of morphological measures on the probability of being found in copula.

Our results, on the other hand, are similar to those obtained in the non-synchronous *P. greeni*, in which the size of lanterns, eyes and body were not related to the probability of males being found alone or in copula (*Demary, Michaelidis & Lewis, 2006*). In this species, as well as in other *Photinus* species (*Branham & Greenfield, 1996*; *Cratsley & Lewis, 2003*; *Demary, Michaelidis & Lewis, 2006*; *Lewis, 2016*), elements of the flashing pattern are important in determining male mating success. However, elements of the flashing pattern are also important in *Photinus ignites*, a non-synchronous firefly in which a significant effect of lantern size and body size on mating success has also been observed (*Cratsley & Lewis, 2003*, *2005*). A study of the effect of the flashing pattern on mating success, and its possible interaction with lantern size in *P. palaciosi* remains as an interesting possibility.

The morphology, physiology and behavior of signal detection and emission organs are frequently influenced by selective pressures not related to the sexual communication function (*Niven & Laughlin, 2008*; *Stöckl et al., 2013*; *Elgar, Johnson & Symonds, 2019*). Thus, another possible explanation for our results is the existence of additional selective pressures acting in opposite direction to sexual selection or in a more complex way. Although the lantern of adult fireflies is an organ for emitting sexual signals, it can be subject to natural selection (*Branham & Wenzel, 2003*; *Woods et al., 2007*; *Lewis & Cratsley, 2008*; *Stanger-Hall et al., 2018*). For example, a study of two *Photinus* species

determined that flashing increases predation risk and metabolic rate (37% with respect to the basal metabolic rate, even though the experimental setting excluded flight) (*Woods et al., 2007*). In *P. palaciosi* it is not known if some predator exerts a similar pressure on signaling fireflies and if this possible effect is related to lantern size. We have made non-systematic observations of several unidentified predators (a grasshopper and species of orb-webb spiders) that capture males during the mating period, although light emission seems irrelevant in prey capture at least for orb-webb spiders.

The steeper allometric slope of lantern size in males suggests that the payoff of investing in lanterns proportional to their size may be higher in large males compared to large females. However, the fact that the slope of females is still higher than 0 suggests that large females also benefit from investing in lanterns somewhat proportional to their size. This is consistent with the finding that there is assortative mating in terms of relative lantern size, suggesting some degree of female-female competition and/or male choice. A likely scenario explaining this pattern may be that among the few females present during a given night, those with larger lanterns could be more detectable to males, or more attractive as the lantern also predicts female size and thus could predict fecundity. In this case, larger females would be the first to mate and, among males competing for these larger females, those with larger lanterns may be detected or selected first by these females. The fact that assortative mating was not found in terms of absolute lantern size but in terms of relative lantern size independently of female body condition is intriguing. At least prior to physical proximity and during the first visual signaling interactions, body size can hardly be assessed by either sex but lantern size could. Relative lantern size may be an honest signal of quality in the case of males as it could show its energetic efficiency independently of body size, or a Fisherian trait, but in the case of females it may be deceiving males if they use it to assess female fecundity as it would not reflect absolute female size or condition, but would still make females with larger lanterns more detectable regardless of their body size. This scenario would not only explain our results in terms of sexual dimorphism, lantern allometry and assortative mating; if males and females throughout the size range of both sexes end up mating, no association between any morphological trait and the probability of being found in copula is expected to arise. Some degree of male mate choice in *Photinus* fireflies has been suggested before (*Lewis, Cratsley & Demaris, 2004*; *Lewis, Cratsley & Rooney, 2004*), especially since the payoff for males of mating with low fecundity females may be negative when a costly nuptial gift is offered, as it is common in this genus. However, although this still unknown for *P. palaciosi*, female flightlessness as been shown to be associated to loss of spermatophore production in males (*South et al., 2011*). We are currently investigating if *P. palaciosi* produces nutritious spermatophores.

Not only lantern size was not related to the probability of being found in copula, neither absolute or relative eye size predicted the probability of being found in copula in either sex. As discussed in the case of lantern size, additional selective pressures affecting eye size and acting in opposite or more complex ways could explain these results (*Lau & Meyer-Rochow, 2006*). For example, the detection and assessment of visual signals of mate quality (*Lewis, 2016*; *Rosenthal, 2017*; *Elgar, Johnson & Symonds, 2019*;

*Stanger-Hall et al., 2018*) suggests that the structure and function of the eyes has evolved influenced by intersexual selection (mate choice). However, the eyes are also used to navigate through the habitat, find other resources (food, shelter, etc.) and detect natural enemies and, thus, its evolution is also affected by natural selection (*Elgar, Johnson & Symonds, 2019*). As mentioned above, we have observed several predators that capture males during the mating period and could be significant selective pressures on eye size. Interestingly, unlike most species of *Photinus*, eye size was smaller in males than in females, suggesting that in females selection pressures derived from processes such as female choice, predator avoidance and the choice of perch for mate location, could be important to understand the evolution of eye size. The high variation and allometric slopes of eye size in both sexes imply that this trait is unrelated to body size throughout the body size ranges of both males and females, suggesting that, unlike lanterns, there is little or no selection for larger males or females to invest in eyes proportional to their size. Although having large lanterns and eyes may represent selective advantages (*Lloyd, 1966*; *Demary, Michaelidis & Lewis, 2006*), a trade-off may restrict the possibility of investing in both functions (structures). It would seem that both sexes favor investing in signaling (lanterns) proportionally to their size, while investment in reception (eyes) is highly variable and independent of body size.

Firefly populations worldwide are declining and threatening factors vary in importance for different species and regions (*Lewis et al., 2020*). Although there is a recent and important interest in firefly watching as a tourist attraction, conservation measures and regulation of touristic activities in fireflies "sanctuaries" need to be based on solid scientific information. Light pollution and tourism are considered important threats for the charismatic synchronous species, such as *P. palaciosi*, and these factors have their main impact during the mating period. Unfortunately, mating dynamics have been studied only in a handful of the about 2,000 firefly species described.

## CONCLUSIONS

Sexual dimorphism in lantern and eye size, along with allometric slopes of lantern size, and assortative mating in terms of relative lantern size, support not only the hypothesis of intense sexual selection among males of *Photinus palaciosi*, but also the possibility of subtle mechanisms of sexual selection among females as well. Trade-offs between investment in signaling (lanterns) versus detection (eyes) structures, or with pressures different from sexual selection such as those imposed by predators, are also likely to be important in shaping the evolution of sexual signaling in this species.

## ACKNOWLEDGEMENTS

We thank the field assistance of Jaime Camacho, David Xochipiltécatl, Mixtli Crisóstomo Pérez and Israel Hernández. We thank the technical help of Raúl Iván Martínez and Sahid Robles. This research is part of the doctoral thesis of Tania López-Palafox in the Posgrado en Ciencias Biomédicas (Universidad Nacional Autónoma de México).

### Funding

This research was supported by a grant from PAPIIT/UNAM (IN219818) to Carlos Cordero. Tania López-Palafox (TLP) is supported by a CONACYT (México) Scholarship. Part of the writing was made during a visit of TLP to Dr. Sara Lewis lab (Tufts University) supported by the program PAEP of the Universidad Nacional Autónoma de México. The funders had no role in study design, data collection and analysis, decision to publish, or preparation of the manuscript.

### Grant Disclosures

The following grant information was disclosed by the authors:
PAPIIT/UNAM: IN219818.
CONACYT (México) Scholarship.
Universidad Nacional Autónoma de México.

### Competing Interests

The authors declare that they have no competing interests.

### Author Contributions

- Tania López-Palafox conceived and designed the experiments, performed the experiments, analyzed the data, prepared figures and/or tables, authored or reviewed drafts of the paper, and approved the final draft.
- Rogelio Macías-Ordóñez conceived and designed the experiments, analyzed the data, prepared figures and/or tables, authored or reviewed drafts of the paper, and approved the final draft.
- Carlos R. Cordero conceived and designed the experiments, analyzed the data, prepared figures and/or tables, authored or reviewed drafts of the paper, and approved the final draft.

### Field Study Permissions

The following information was supplied relating to field study approvals (i.e., approving body and any reference numbers):

Our field collection was made under the Secretaría de Medio Ambiente y Recursos Naturales (SEMARNAT, Mexican Government) permit SGPA/DGVS/06292/16.

### Data Availability

Raw data and script are available in the Supplemental Files.

### Supplemental Information

Supplemental information for this article can be found online at http://dx.doi.org/10.7717/peerj.10127#supplemental-information.

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
