# Peer review of "The size of signal detection and emission organs in a synchronous firefly: sexual dimorphism, allometry and assortative mating"

_PeerJ, doi:10.7717/peerj.10127_

## Round 0.1 · original submission · Major Revisions

Dear Drs. López-Palafox and Cordero:

Thanks for submitting your manuscript to PeerJ. I have now received three independent reviews of your work, and as you will see, one reviewer recommended rejection, while another suggested major revision (with many suggested changes). I am affording you the option of revising your manuscript according to all three reviews.

The reviewers generally agree that your manuscript raises interesting questions about sexual dimorphism and mate choice. The novelty of your work needs to be better delivered in the light of previous studies. It is perceived that mate choice data are weak and sample sizes are too small. Data were collected from different localities that were often several kilometers apart. As these fireflies are not known to be long-distance fliers, and males are known to live only a short period, each capture may represent different populations with no replications. Please address this.

Please address Reviewer 2’s concern that “the validity of the findings largely hinges on the appropriateness of eye size as the proxy for ability to detect point light sources. This is where the authors need to deliver more information and justification”.

Please ensure that all R scripts are available to allow replication of your analyses. All aspects of your work should be repeatable. It appears that certain key references are missing. Please add these, as it seems they may help in better framing your hypothesis.

The reviewers raised many minor concerns about the manuscript, as well as constructive suggestions. Please address all of these issues.

I look forward to seeing your revision, and thanks again for submitting your work to PeerJ.

Good luck with your revision,

-joe

Reviewer 1 ·

Basic reporting

L53-55, please specify if these species have synchronous or asynchronous males

L59, I agree that synchronous species are good subjects due to the density of signaling males. But it would be helpful to discuss the downside which is that males with a smaller lantern might be able to exploit larger neighboring males without paying the severe metabolic costs. Also at line 62 aren’t the authors assuming that there is no pre-copulatory courtship or cryptic female choice by the females, as discussed by Eberhard and others?

L72, change for to of

L82, at first I agreed that selection should favor males with larger eyes for improved detection ability but then I wondered more about the natural history of how far a male can or will fly to try to encounter female: larger eyes should capture more photons meaning a male could detect a female at a greater distance. However under intense competition, by the time he actually arrives, another male closer by may have already gotten there. In synchronous species presumably females are not responding to just one particular male, but to the collective source of light (followed by scramble competition among males to get there first). If so, males be selected to respond via controlled flight to point sources only within a certain range, which may be a constraint on eye size enlargement.

I wonder if the situation might be similar to some deep-sea fishes that, if I recall correctly, have relatively smaller eyes, which is hypothesized to be associated with being relatively weak swimmers (see Warrant and colleagues, Biol Rev 2004). It may be unfair to claim that fireflies are relatively weak flyers but I think the argument is at least worth considering, as it would flip the prediction the authors make.

L121, it was not clear to me how female body size was estimated given size was based on an estimate of elytra length IF the elytra were not reduced, but how can this be known?

L125ff, it would be useful to show that there are, or are not, allometric relationships between body size and eye size, and lantern size.

L154, I am not persuaded these data showing female elytra are shorter than those of males are worth presenting since we already know they are brachyopterous.

L190, in discussing support for their hypothesis, the authors seemingly jump from intra-specific comparisons to inter-specific comparisons. That is, they set up to study by predicting that males with larger eyes and or larger lanterns should be found in copula than males with smaller structures. They found no evidence for this. But then these inferences really are inter-specific, showing that males are larger bodied, and have larger lanterns, than females. To fully substantiate the claim that it is due to sexual selection requires out-group comparisons, showing that the derived species have larger lanterns, for example. This should be clarified (similarly, for female eye size line 194).

L197, here the authors need to again clarify which species are synchronous and which are asynchronous, as I think the predictions differ.

L204, as above is this species synchronous or asynchronous?

L211, the observation that most meetings involve solitary males contradicts the notion of intense competitiveness described in line 205. Please reconcile.

L228, correct spelling of species name

L 233, fix typo in Laughlin

L239, change webbing to web (also L 252)

Experimental design

No comments in addition to the above

Validity of the findings

No comments apart from the above

Additional comments

I enjoyed reading this manuscript very much, and found the study well done, interesting and compelling.

Reviewer 2 ·

Basic reporting

The authors use clear and unambiguous, professional English throughout their work and provide literature that is mostly sufficient to support their statements. However, I have noticed gaps concerning their cited literature works in support of setting up their study and formulating their hypotheses which may potentially question their justification of the study. Please see my comments below, “Experimental Design.” The structure of the article is professional, and raw data are shared. However, R scripts to replicate their analysis are not supplied which is a shortcoming that must be addressed.

Experimental design

The paper represents original research within the aims and scopes of the journal. The authors identify a gap of knowledge about firefly mating systems and underlying mechanism, but the main research question is not well defined and perhaps may not even be meaningful.
The authors must explain how eye size is linked to detection ability of bioluminescent light flashes. Following Land and Warrant’s extensive body of work on physiological modeling of visual light sensitivity, it is primarily the absolute size of the optical aperture which determines the ability of the eye to detect point light sources. Of course, the size of the aperture will be constrained by the size of the eye and animal, hence one would need to look at absolute aperture size and relative aperture size. Eye size, as used by the authors, may not be meaningful in the detection ability of point light sources such as bioluminescent light flashes. In addition, the authors must justify the use of lantern size as a proxy of emitted light intensity of the signal and provide evidence that the size of the lantern organ is indeed positively correlated with signal intensity.
As for the methods, I notice that R scripts were not supplied, which is not conducive to reproducing the statistical analyses. The authors should share their scripts for full transparency. In addition, the chosen methods are far from ideal. Methods should explicitly address body size as a covariate, hence the analyses should focus on ANCOVAS. ANCOVAS are much more “direct” than PCA and functionally more relevant, because the loadings of the PC represent a mix of the traits. It is therefore difficult to explicitly ascribe a function to the loadings. ANCOVAS also allow for using scaling information as an assessment of selection, such as isometry vs allometry.

Validity of the findings

The validity of the findings largely hinges on the appropriateness of eye size as the proxy for ability to detect point light sources. This is where the authors need to deliver more information and justification.

Reviewer 3 ·

Basic reporting

Basic reporting is sound and clear and paper is well written. Publishing negative results is important if they are based on reliable data and scientifically sound hypotheses. In the present state, the paper does not reach scientific standards. This paper could become scientifically sound after rewriting and reanalyzing the data.
A) Authors should be aware of the hypotheses proposed to explain sexual size dimorphism in Lampyrid species (relevant literature listed below).
B) Correlational data are based on samples captured from different field populations each collected in different times. The data are relevant to describe sexual dimorphism of the species but not relevant to test hypothesis of mate choice. The data also have a limited value as correlation is not causation.
C) The description of mating system of Photinus palaciosi is lacking.

Detailed comments:
Sexual dimorphism and mating system
1.There are several published hypotheses explaining sexual dimorphism in Lampyridae. Male biased sex ratio is not the only explanation to a large male (and lanterns) size. In many species (flightless) females are larger than males and yet male-male competition is intense (see below).
Sexual size dimorphism in fireflies has been connected to female flightlessness, evolution of female sexual signals and male nuptial gifts (South et al., 2020, see below).
Is P. palaciosi a nuptial a gift giving species?
Most firefly species are capital breeders and adults do not feed. Thus both females and males have a limited energy for reproduction and maintenance. For example, some Photinus species females may loose seven eggs/day if remain unmated (Wing, 1989).
During mating males of many Photinus species deliver a large spermatophore which increases female fecundity and longevity (Lewis et al., 2011). Male nuptial gift size correlates positively with body size. A large nuptial gift delays female remating and increases his fertilization success. A large nuptial gift also increases female fecundity and longevity. Thus large male size may be selected not because of lantern size but because of spermatophore size.
Is this the case in P. palaciosi as well?
Thus testing correlation between lantern size (signal emission organ) with mating success may be irrelevant.
Large male size may be selected because large spermatophores delay female rematings and increase her fecundity and thus his fertilization success. This should be tested in controlled experiment and cannot be detected in field collected data.

References not found in the ms:
Lewis, S. M., Cratsley, C.K., Rooney. J. A., 2004. Nuptial gifts and
sexual selection in Photinus fireflies. Integr. Comp. Biol. 44:234–237.

South, A., Stanger-Hall, K., Jeng, M.-L., Lewis, S.M., 2011. Correlated evolution of
female neoteny and flightlessness with male spermatophore production in
fireflies (Coleoptera: Lampyridae). Evolution 65, 1099–1113.

Wing, S.R., 1989. Energetic costs of mating in a flightless female firefly, Photinus
collustrans (Coleoptera: Lampyridae). J. Insect Behav. 2:841–847.

2. Male-biased sex ratio
Authors claim “a strong male-biased sex ratio” (line 23). In their own data male biased sex ratio was found in all eight early samples/localities but in nine late samples sex ratio was males biased only in two samples (22% of samples): it was female biased in three samples and equal in four sampels, see ms: supplementary material).
In other Photinus species sex ratio is known to vary temporally. Local sex ratios become female-biased later in mating season (Lewis & Wang,1991). Could that be the case also in P. palaciosi?
References not cited in ms:
Lewis, S. M., and Wang, O. T. 1991. Reproductive ecology of two species of Photinus fireflies (Coleoptera: Lampyridae). Psyche 98:293–307.

Experimental design

The work is based on correlative data from field captures. The data are collected from different localities and different dates. The distance between sampling localities are not presented in supplementary material. There is seventeen days between the first and the last sampling. To pool the data from different samples is valid when testing sexual dimorphism in morphological measurements.
When testing possible mate choice hypotheses pooled data across sites and localities are not valid. For example, some males (altogether 34) were captured in localities (and on a period) when no females were found. Thus these males could not make any choice or could not be a target for female choice. The data should be reanalysed based only on samples when both solitary and mating individuals are found simultaneously and date (= locality) should be one explanatory factor in the analysis. Figure 2 and Table 4 should be rewritten respectively.
The relationship between signal detection organ (eyes) and lantern size with mating success should be tested experimentally not sampling field data. The effect of male nuptial gift on male mating success should also be tested.

Validity of the findings

When the data are reanalysed, irrespective of the results, the findings may be interesting if they are discussed on the light of mating systems of firefly species (South et al. 2011). Even when reanalysis is done, the present data are correlative and not evidence of causality.

Additional comments

The paper gives correlative data based on field samples. The paper is well written and clear but currently lack strong scientific merits. The authors should include more information on mating system of the species. Sexual size dimorphism should be discussed and tested on the light of published hypotheses. The data on mate choice should be reanalyzed omitting localities where only individuals of one sex are found.

---

## Round 0.2 · accepted · Accept

Dear Drs. López-Palafox and colleagues:

Thanks for revising your manuscript based on the concerns raised by the reviewers. I now believe that your manuscript is suitable for publication. Congratulations! I look forward to seeing this work in print, and I anticipate it being an important resource for groups studying firefly biology and mating behavior. Thanks again for choosing PeerJ to publish such important work.

Best,

-joe

Reviewer 3 ·

Basic reporting

I feel there is no need to re-review. The ms has been rewritten well according to suggestions.

Experimental design

.

Validity of the findings

.